# Associations between Emotion Regulation, Feeding Practices, and Preschoolers’ Food Consumption

**DOI:** 10.3390/nu14194184

**Published:** 2022-10-08

**Authors:** Ana Filipa Santos, Carla Fernandes, Marília Fernandes, António J. Santos, Manuela Veríssimo

**Affiliations:** William James Center for Research, ISPA-Instituto Universitário, 1149-041 Lisboa, Portugal

**Keywords:** emotion regulation, feeding practices, food consumption, preschool

## Abstract

Previous research identified emotion dysregulation, non-responsive feeding practices, and unhealthy food consumption as risk factors for childhood obesity. However, little is known about the relationships between these factors. This study examined associations between children’s emotion regulation, parental feeding practices, and children’s food consumption. The sample consisted of 163 mothers of children aged 3–5 years. Mothers completed the Emotion Regulation Checklist, the Child Feeding Questionnaire, and the Child Health Section from the Parent Interview of the Early Childhood Longitudinal Study-B to assess model variables. Results showed that healthy food consumption was associated with higher emotion regulation abilities, higher monitoring, and lower pressure to eat. For unhealthy food consumption, the associations were in opposite directions. Higher emotion regulation abilities were also associated with higher monitoring, lower pressure to eat, and lower restriction. For lability, the associations were in opposite directions. Regression analyses revealed that children’s lability, pressure to eat, and monitoring were significant predictors of children’s food consumption. These findings suggest that children’s emotion regulation and feeding practices are important determinants of children’s food consumption. Future longitudinal studies that examine bidirectional associations between children’s emotion regulation, parental feeding practices, children’s food consumption, and potential mechanisms accounting for these associations are needed.

## 1. Introduction

Childhood obesity is a serious global health problem with a significant impact on physical and psychological health in both short-term and long-term periods [1,2,3,4]. Children who are overweight or obese have a higher risk for premature death and disability in adulthood [5,6,7] and are more likely to become overweight and obese adults [8,9].

Eating habits and food preferences are established early in life and carried into adulthood [10,11,12,13]. Obesity begins during preschool years; thus, this period is crucial for the development of healthy habits, namely healthy food consumption [14]. Healthy food consumption, which is characterized by a higher intake of fruits and vegetables, is a protective factor against obesity [15,16]. In reverse, unhealthy food consumption, described as a higher intake of energy-dense, sugary, low-quality foods, is associated with an increased risk of obesity [16,17]. Parental feeding practices directly contribute to childhood obesity and have an important role in children’s food consumption (e.g., [18,19,20,21]). Thus, it is important to consider these behaviors/strategies.

### 1.1. Feeding Practices and Food Consumption

Infants and young children are initially dependent on parents for food, so they are the main food providers and the most powerful socialization agents in children’s eating behavior, as they select the foods children will eat and serve as models of eating [22,23]. Therefore, parental feeding practices play a crucial role in shaping children’s eating behaviors, the self-regulation of energy intake, and consequently children’s weight [24,25,26,27,28,29,30]. Parental feeding practices can be considered responsive or non-responsive [31,32,33]; responsive feeding practices reflect parents’ ability to accurately recognize and respond to children’s cues of hunger and satiety, whereas non-responsive feeding practices are incongruous with these cues and may undermine children’s self-regulation of energy intake, placing them at risk for excessive weight gain [26,29,30,31,34,35,36,37].

The two most studied non-responsive feeding practices are restriction and pressure to eat [27,38,39,40,41]. Pressure to eat refers to the extent to which parents enforce or strongly encourage their children to eat more food or specific foods bypassing their satiety cues, particularly at mealtimes [23,38,42,43]. Restriction involves parents limiting children’s consumption of food, namely palatable energy-dense foods, even when the child is hungry [23,42,44]. These practices, although well-intended, can lead to outcomes that are opposite of the parents’ intentions and be counterproductive [31,39,45]. In this sense, the pressure to eat has been associated to a lower consumption of fruits and vegetables and a higher consumption of high-fat foods, as pressuring children to eat particular foods relates to a decreased preference for and consumption of these foods [27,46,47]. On the other hand, restricting foods can lead to an increased preference for and the consumption of the restricted foods once it is available and overeating in the absence of hunger [26,40,48,49,50,51].

Responsive feeding practices are less studied, but evidence suggests that they are essential for the development of healthy eating habits [52,53,54]. In particular, parental monitoring, which is the practice of keeping track of one’s child consumption (namely, high far or sugary foods), has been associated with a healthier food consumption [49,55,56]. Specifically, monitoring is associated with children’s consumption of fewer unhealthy foods, such as ultra-processed food and sugary drinks and with children’s higher consumption of healthy foods, such as fruits, vegetables, and high-fiber foods [54,56,57,58,59]. Consequently, the feeding practices that parents choose to use are significant determinants of children’s food consumption.

### 1.2. Emotion Regulation and Food Consumption

An increasing body of research shows that children’s self-regulation plays an important role in the development of childhood obesity (e.g., [60,61,62,63,64]). One component of self-regulation, emotion regulation, has been implicated in the development of obesity in early childhood. More specifically, poor emotion regulation in children has been associated with a higher risk of becoming overweight or obese [62,65,66].

In fact, we know that children are born with the ability to self-regulate their intake of food [67,68,69,70,71,72] and it is believed that the self-regulation of energy intake operates at an automatic level in terms of starting and stopping eating in response to hunger and satiety cues [61,73]. However, this ability seems to decrease with age [74,75,76,77,78]. That is because, besides innate predispositions, children’s self-regulation of energy intake is not only also shaped by external factors, namely, as previously mentioned, parental feeding practices, but also by stressful/negative early experiences and emotional factors [61,62,63,64,65,66,67,68,69,70,71,72,73,74,75,76,77,78,79,80]. Indeed, as supported by Hamburg and colleagues [81], emotional states can influence when, how much, and what people eat, and in turn, the food consumed affects subsequent emotional states. In fact, food, especially those rich in calories, produces a calming effect in brain areas involved in stress responses [82,83,84]. Consequently, children with lower emotion regulation abilities may have difficulties attenuating their negative emotions and my resort to food (namely, palatable food) to cope with their distress, a response known as emotional eating [83,85,86,87]. Emotional eating has been associated with increased consumption of high-fat and sugary food and a decreased consumption of fruits and vegetables [88,89,90,91].

Preschool years are critical for the development of emotion regulation [92,93]; since emotion regulation difficulties can lead to emotional eating [94,95], which, in turn, is related to unhealthy food consumption, it is important to investigate the relationship between emotion regulation and food consumption during this period. A small group of studies studied this relationship in school-aged children and adolescents, with superior emotion regulation skills being associated with healthy food consumption [96,97,98,99]. However, little is known about this relationship in preschoolers.

### 1.3. Current Study

There is a dearth of literature linking children’s emotion regulation, parental feeding practices, and children’s food consumption. The majority of studies connecting emotion regulation to feeding practices focused mainly on emotion regulation feeding practices (that is, parents’ use of food to regulate children’s emotional arousal) [100,101]. As previously mentioned, there is also a dearth of data on the relation between emotion regulation and food consumption in preschool years. Thus, the aim of the current study was to explore the relation between preschoolers’ emotion regulation, parental feeding practices (specifically, pressure to eat, restriction, and monitoring), and preschoolers’ food consumption, reported by mothers.

## 2. Materials and Methods

### 2.1. Participants

Participants were mothers of 163 children (41.7% girls and 58.3% boys) between 3 and 5 years old (*M* = 3.90; *SD* = 0.88): 62.6% were firstborns, and 68.1% had siblings. Children spend between 3 and 10 h at preschool (*M* = 7.51, *SD* = 1.17). The mothers’ age ranged between 23 and 48 years (*M* = 36.75; *SD* = 4.94), and the fathers’ ranged between 22 and 58 years (*M* = 38.82; *SD* = 5.87). Education levels ranged for mothers from 12 to 21 years (*M* = 15.99; *SD* = 2.77) and for fathers from 12 to 21 years (*M* = 14.59; *SD* = 3.06). The majority of parents worked full-time (mothers 91.4% and fathers 98.1%) and were either married or cohabiting (67.3%).

### 2.2. Measures

#### 2.2.1. Child Emotion Regulation

Children’s emotion regulation was assessed using the Emotion Regulation Checklist (ERC) [102]. ERC consists of 24 items answered on a 4-point rating scale (1 = rarely or never; 2 = sometimes; 3 = often; 4 = almost always), which evaluates caregivers’ perceptions of children’s emotionality and regulation. ERC comprises two subscales. The *Emotion Regulation subscale* (8 items; e.g., “Can say when he/she is feeling sad‚ angry‚ or mad‚ fearful or afraid”) measures adaptive regulation, including appropriate emotional display, emotion understanding, and empathy. Higher scores on the *Emotion Regulation* subscale indicate higher levels of emotion regulation skills. The *Lability/Negativity* subscale (15 items; e.g., “Is prone to angry outbursts/tantrums easily”) measures mood swings, angry reactivity, and intensity of positive and negative emotions. Higher scores on the *Lability/Negativity* subscale indicate higher levels of emotion dysregulation, inflexibility, mood lability, and negative effects. For this study, internal consistencies of the *Emotion Regulation* and *Lability/Negativity* subscales were 0.57 and 0.73, respectively. The *Emotion Regulation Global* score was also used, and the internal consistency was 0.74.

#### 2.2.2. Parental Feeding Practices

The subscales *Pressure to Eat*, *Monitoring*, and *Restriction* from the Child Feeding Questionnaire (CFQ) [44] were used to measure parental feeding practices. *Pressure to eat* (*α* = 0.74) consists of four items (e.g., “My child should always eat all of the food on her plate”), *Monitoring* (*α* = 0.86) consists of three items (e.g., “How much do you keep track of the high fat foods that your child eats?”), and *Restriction* consists of (*α* = 0.71) eight items (e.g., “I have to be sure that my child does not eat too much of his/her favorite foods”). Using a 5-point rating scale, mothers were asked to indicate the degree to which they agreed with each statement (1 = disagree; 2 = slightly disagree; 3 = neutral; 4 = slightly agree; 5 = agree) or how often they used a feeding practice (1 = never; 2 = rarely; 3 = sometimes; 4 = mostly; 5 = always).

#### 2.2.3. Child Food Consumption

Children’s food consumption was assessed using the Child Health Section from the Parent Interview of the Early Childhood Longitudinal Study, Birth Cohort (ECLS-B) [103]. Mothers were asked to report the frequency with which the child consumed specified foods or drinks over the seven previous days (e.g., “During the past seven days, how many times did your child eat vegetables other than French fries and other fried potatoes?”). Response options are as follows: “once a day”, “twice a day”, “three times a day”, “four or more times a day, “one to three times during the past seven days”, “four to six times during the past seven days, and “the child did not eat/drink [item] during the past seven days”. Mothers could also respond with “refused” or “don’t know”. As proposed by Bost and colleagues [104], composite scores were created to reflect total healthy food consumption (average of the “fresh fruit” and “vegetables” items) and unhealthful food consumption (average of the “sugary drinks,” “salty snacks,” “candy and sweets,” and “fast food” items).

### 2.3. Procedures

This study is part of the wider research project “ChildObesity—Child obesity risk: the role of attachment, child’s temperament and self-regulation”, approved by the Ethics Committee of the ISPA—Instituto Universitário. Participants were recruited from public schools in the Lisbon Metropolitan Area. The study’ goals and procedures were presented to school boards to obtain authorizations for data collection. Informed consent and questionnaires were sent to mothers through their children’s teachers and returned to in a sealed envelope after completion.

### 2.4. Analytic Plan

All statistics were run using the Statistical Package for the Social Sciences (SPSS, Version 28.0, Armonk, NY, USA). Before our main analyses, descriptive statistics were explored using the mothers’ reports. Mean differences, for categorical variables, were explored using independent *t*-tests, and Pearson’s correlation was used to explore the associations with continuous variables. The mean differences between the instruments’ dimensions were explored using the pair samples *t*-test. Relations between mothers’ ERC, CFQ, and ECLS-B reports were tested using Pearson’s correlation coefficient. Two multiple regressions (one for healthy and other for unhealthy food consumption) were performed to explore the child’s emotion regulation and parental feeding practices contributions to the children’s food consumption.

## 3. Results

### 3.1. Preliminary Analyses

Descriptive statistics are presented in Table 1.

In general, mothers described children as presenting low *Lability* and high *Emotion Regulation Global* scores (*Emotion Regulation* dimension was not used due to the low alpha presented). Parents’ education level was positively associated with mothers’ reports on children’ *Emotion Regulation* (mother *r* = 0.27, *p* < 0.001 and father *r* = 0.27, *p* < 0.001) and negatively with *Lability* (mother *r* = −0.16, *p* < 0.05 and father *r* = −0.18, *p* < 0.05). No other significant differences or correlations were found.

Mothers reported using more *Monitoring* (*M* = 4.30, *SD* = 0.80) than *Restriction* (*M* = 3.02, *SD* = 0.72) or *Pressure to Eat* practices (*M* = 2.63, *SD* = 1.02) (*t* = 18.36, *p* < 0.001 and *t* = 16.58, *p* < 0.001 respectively). Moreover, there were more *Restriction* practices than *Pressure to Eat* practices (*t* = 4.79, *p* < 0.001). Mothers with fewer years of education reported higher *Pressure to Eat* practices (*r* = −0.19, *p* < 0.001). No other significant correlations or differences were found.

Regarding children’s food consumption, mother’s report more healthy than unhealthy scores (*M* = 4.44, *SD* = 1.17, and *M* = 2.15, *SD* = 0.99, respectively, *t* = 14.93, *p* < 0.001). When the child was the firstborn, mothers reported lower unhealthy food consumption scores (*M* = 1.93, *SD* = 0.88, and *M* = 2.52, *SD* = 1.06, for not first-born children, robust *F* = 13.18, *p* < 0.001). Parental education was also significantly associated with food consumption. There was a positive association between mothers’ education and the reports on child healthy food consumption (*r* = 0.35, *p* < 0.001), and unhealthy food consumption reports were also negatively associated with parental education (mother *r* = −0.31, *p* < 0.001, and father *r* = −0.23, *p* < 0.01).

### 3.2. Associations between Children’s Emotion Regulation, Parental Feeding Practices, and Food Consumptions

Healthy food consumption was positively related with children’s *Emotion Regulation* and *Monitoring* feeding practices and negatively with *Pressure to Eat*. For unhealthy food consumption, the relations were in opposite directions (see Table 2).

There was a negative correlation between children’s *Emotion Regulation* and both *Restriction* and *Pressure to Eat* feeding practices and a positive relation with *Monitoring*. For *Lability*, the relations were in opposite directions (see Table 2).

### 3.3. Regression Analyses for Food Consumption Models

In the following analysis, only *Lability* was used as it was highly correlated with the *Emotional Regulation Global* score (*r* = −0.87; *p* < 0.001). Regarding feeding practices, only *Pressure to Eat* (*β* = −0.29; *p* < 0.001) and *Monitoring* (*β* = 0.20; *p* < 0.05) were predictors of the child’s healthy food consumption. For unhealthy food consumption, predictors were also *Pressure to Eat* (*β* = 0.28; *p* < 0.001) and *Monitoring* (*β* = −0.25; *p* < 0.01) (see Table 3).

## 4. Discussion

The aim of this study was to investigate associations between emotion regulation, parental feeding practices, and children’s food consumption. Findings suggest that both children’s emotion regulation and mothers’ feeding practices play an important role in influencing children’s food consumption.

Our correlational analyses revealed that children’s healthy food consumption was associated with mothers’ higher use of monitoring and a lower use of pressure to eat, while unhealthy food consumption was associated with mothers’ lower use of monitoring and higher use of pressure to eat. These findings are in line with previous studies (e.g., [27,46,47,49,56,79]), suggesting that responsive feeding practices may contribute to children’s healthy food consumption, whereas, in reverse, highly controlling/non-responsive feeding practices might be a risk factor for children’s unhealthy food consumption. However, contrary to previous evidence (e.g., [26,40,48,50]), we found no associations between mothers’ use of restriction and children’s food consumption. One possible explanation for this inconsistency could be the lack of distinction between restriction regarding food quality and restriction with regard to food amount in this study [105,106]. In fact, the majority of the CFQ restriction subscale items refer to food quality. Thus, future studies should further investigate different reasons why parents restrict foods (i.e., restriction for health and restriction for weight) [53].

Regarding the relation between children’s emotion regulation and food consumption, our results showed that children’s improved emotion regulation abilities were associated with children’s higher healthy food consumption and lower unhealthy food consumption. These results are consistent with a small group of previous studies [96,97,98,99] and further extended this research by examining this relation for the first time in a preschool sample. Children’s improved emotion regulation abilities were also associated with mothers’ higher use of monitoring and a lower use of pressure to eat and restriction. For children’s higher emotion dysregulation, the associations were in opposite directions. To our knowledge, this is the first study analyzing the association between children’s emotion regulation and feeding practices such as pressure to eat, restriction, and monitoring. Most studies have focused on the connection between parental feeding practices and children’s emotional eating (e.g., [107,108]), revealing that the pressure to eat and restriction were associated with children’s emotional eating. As emotion regulation difficulties could be a risk factor for emotional eating [85,94,95], it would be interesting for future studies to assess children’s emotion regulation as a possible moderator that may strengthen relationships between parental feeding practices and emotional eating. 

Regression analyses demonstrated that children’s emotion regulation and feeding practices were significant predictors of children’s food consumption. More specifically, children’s higher emotion dysregulation, higher pressure to eat, and lower monitoring predicted unhealthy food consumption. On the other hand, children’s lower emotion dysregulation, lower pressure to eat, and higher monitoring predicted healthy food consumption. A closer examination of this finding could suggest that monitoring could be a protective factor for unhealthy food consumption. Together, these results suggest that children’s food consumption may be determined by children’s emotion regulation and parental feeding practices, with lower emotion regulation competence and non-responsive feeding practices translating into unhealthier food consumption and, in contrast, more emotion regulation competence and responsive feeding practices translating into healthier food consumption. These findings also highlight an important question for future research: What are the different mechanisms and pathways through which emotion regulation and feeding practices influence children’s food consumption? For instance, children’s emotion regulation could be related to feeding practices through parental emotional responsiveness. As caregivers strongly influence children’s self-regulation of energy intake through their feeding practices, they also play a powerful role in influencing children’s self-regulation of emotions through the ways they respond to their emotions [29,79,109]. In this regard, a growing body of research suggests that emotional responsiveness parallels feeding responsiveness with the use of unsupportive emotional responses (e.g., punitive, minimization, and distress responses), setting the stage for the use of non-responsive feeding practices (e.g., pressure to eat, food as reward, and emotion regulation) [32,33,79,110]. Therefore, the manner through which caregivers respond to their children’s emotions around food can not only impact children’s emotion regulation but also children’s regulation of energy intake. Bost and colleagues [104] found further evidence using a serial mediation model, revealing associations between attachment security, emotional responsiveness, feeding responsiveness, and food consumption. They discovered that insecure mothers were more likely to use unsupportive responses to children’s negative emotions, which increased the likelihood that they would use non-responsive feeding practices; in turn, these feeding practices predicted children’s unhealthy food consumption. More studies accounting for these variables are needed to test these relations and the direction of causal influences.

### Limitations

The limitations of this study should be acknowledged. First, we used cross-sectional data, which precludes the examination of causal relationships. Future research should consider using longitudinal designs to better understand the complex interactions between parents and children around food, as previous evidence suggests that these interactions are bidirectional, with parental feeding practices influencing but also responding to children’s eating behaviors, weight, and characteristics such as temperament and self-regulation [79,111,112,113,114]. Moreover, only maternal self-report measures were used, leaving room for response bias and precluding the generalization of our results to other caregivers. Thus, a multi-informant approach and observational measures are needed to provide a more comprehensive picture of the parent-child feeding dynamic, especially because caregivers may underestimate their use of specific feeding practices and be unaware of the ways they influence children’s food consumption [79]. Lastly, as previously mentioned, other variables might be responsible for connections between children’s emotion regulation, feeding practices, and children’s food consumption. In this sense, more research is needed to identify mechanisms through which emotion regulation and feeding practices may translate into children’s food consumption to advance our understanding of early pathways to obesity risk.

## 5. Conclusions

Despite these limitations, to our knowledge, this is one of the first studies that examines the relation between emotion regulation, parental feeding practices, and children’s food consumption. Considering that obesity begins in preschool and that, once established, the condition is difficult to reverse [14], it is necessary to investigate potential psychological factors that can influence its development during early ages and understand the complex relationships between them. Our findings suggest that children’s emotion regulation and parental feeding practices are both important for understanding developmental pathways relative to young children’ eating behaviors and obesity risk. By better understanding the conditions that influence the links between children’s emotion regulation, feeding practices, and children’s food consumption, we can help providers and researchers develop effective prevention and intervention programs that assist parents in creating appropriate eating environments and foster healthy eating patterns.

## Figures and Tables

**Table 1 nutrients-14-04184-t001:** Means and standard deviations regarding ERC, CFQ, and ECLS-B.

		Mother (*n* = 163)	
		*M*	*SD*
ERC	Emotion Regulation	3.52	0.34
	Lability	1.82	0.34
	Emotion Regulation Global	3.32	0.28
CFQ	Pressure to Eat	2.63	1.02
	Monitoring	4.30	0.80
	Restriction	3.02	0.72
ECLS-B	Healthy Food Consumption	4.44	1.17
	Unhealthy Food Consumption	2.15	0.99

*M* = mean; *SD* = standard deviation; ERC = Emotion Regulation Checklist; CFQ = Child Feeding Questionnaire; ECLS-B = food consumption assessed with the Child Health Section from the Parent Interview of the Early Childhood Longitudinal Study.

**Table 2 nutrients-14-04184-t002:** Associations between ERC, CFQ, and ECLS-B dimensions.

		CFQ			ECLS-B	
		Restriction	Pressure to Eat	Monitoring	Healthy	Unhealthy
ECR	Lability	0.27 ***	0.18 *	−0.17 *	0.00	0.11
	Emotion Regulation Global	−0.24 ***	−0.16 *	0.26 ***	0.16 *	−0.21 **
CFQ	Restriction				0.02	0.10
	Pressure to Eat				−0.26 ***	0.31 ***
	Monitoring				0.19 *	−0.21 **

* *p* < 0.05; ** *p* < 0.01; *** *p* < 0.001.

**Table 3 nutrients-14-04184-t003:** Regression coefficients of child emotion regulation and parenting feeding practices for both healthy and unhealthy food consumption.

Variable	Healthy Food Consumption			Unhealthy Food Consumption		
	B	*β*	SE	B	*β*	SE
Constant	3.41		0.74	2.44		0.62
Lability	0.26	0.08	0.28	−0.03	−0.01	0.23
Restriction	0.06	0.04	0.14	0.12	0.09	0.12
Pressure to Eat	−0.33	−0.29 ***	0.09	0.28	0.28 ***	0.08
Monitoring	0.29	0.20 *	0.12	−0.31	−0.25 **	0.10
R^2^	0.11			0.15		

* *p* < 0.05; ** *p* < 0.01; *** *p* < 0.001.

## Data Availability

The raw data supporting the conclusions of this article will be made available by the authors, without undue reservation.

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
