# Peer review of "Associations between Emotion Regulation, Feeding Practices, and Preschoolers’ Food Consumption"

_nutrients, 2022, doi:10.3390/nu14194184_

Round 1

Reviewer 1 Report

Dear Authors,

The manuscript (nutrients-1895624) submitted for review is interesting. Despite the positive impression this manuscript made on me, I have a few comments.

Comments and Suggestions for Authors

 A title: The title seems to fit the experimental work developed.

Abstract: The abstract is generally adequate, but needs to be reformulated because it is too general. The choice of keywords was appropriate.

 Introduction: The introduction correctly helps to frame and contextualize the work.

Lines 38-39: “Parental feeding practices directly contribute to childhood obesity and have an important role in children’s food consumption  (e.g., [18–21])”. I don’t understand why the authors used brackets and "e.g." in this sentence. Is text missing here?

The same situation is in Discussion: “However, contrary to previous evidence (e.g., [26,40,48,50]), we found no association between mothers’ use of restriction and children’s food consumption”. (lines 240-241).

Lines 55-58: Restriction involves parents limiting children’s consumption of food, namely palatable energy-dense foods, even when the child is hungry [23,42,44]. These practices, although well-intended, can lead to outcomes that are opposite of the parent's intentions and be counterproductive [31,39,45]”. And in lines 61-63, there is a continuing this thought “On the other hand, restricting foods can lead to increased preference for and consumption of the restricted food once it is available and overeating in the absence of hunger [26,40,48–51].”  In my opinion text from lines, 61-63 should be joined with text in lines 55-58.

 Materials and methods:

Lines 114-115: Authors wrote “Children spend between 3 to 10 hours at school (M = 7.51, SD = 1.17).” Is it correct, because the manuscript is related to children in preschool (in kindergarten?)?

Lines 118-120: Authors wrote: “The majority of parents worked full-time (mothers 91.4% and fathers 98.1%) and were either married or cohabiting (67.3%).” What about other parents (32.7%)? Are they lonely?

What statistical program was used to analyze the data?

Results

Lines 180-182: Authors wrote: “This section may be divided by subheadings. It should provide a concise and precise description of the experimental results, their interpretation, as well as the experimental  conclusions that can be drawn.” In my opinion, these sentences are not necessary.

 Authors should add separate sections for limitations and conclusions.

References

References are cited according to journal rules. Positions of references such as 9,24,28,40,67,68, 69,70,71,72,102,109 are old and come from before 2000. Are they necessary?

Reference 104: Bost, K.K.; Wiley, A.R.; Fiese, B.; Hammons, A.; McBride, B.; SRONG KIDS Team. Associations between adult attachment style, emotion regulation, and preschool children’s food consumption. J. Dev. Behav. Pediatr. 2014, 35, 50–61. https://doi.org/10.1097/01.DBP.0000439103.29889.18.

In my opinion, it should be ‘STRONG KIDS’.

Despite my comments, I am pleased to recommend this manuscript for publication.

Reviewer

Reviewer 2 Report

My overall impression of the manuscript is positive, I have suggestions for the Authors to correct the description of statistical analyses and their results.

My main concern is about the regression analyses - I am not sure what model was tested with which analysis. Please describe it more carefully and more clearly, perhaps add a table with all predictors/dependent variables listed. and put all effects into it.

Minor remark - please check the manuscript for unnecessary text, e.g. in the results section, at the beggining.
